# Intraoral Ultrasound versus MRI for Depth of Invasion Measurement in Oral Tongue Squamous Cell Carcinoma: A Prospective Diagnostic Accuracy Study

**DOI:** 10.3390/cancers16030637

**Published:** 2024-02-01

**Authors:** Mikkel Kaltoft, Christoffer Holst Hahn, Marcus Wessman, Martin Lundsgaard Hansen, Tina Klitmøller Agander, Fatemeh Makouei, Irene Wessel, Tobias Todsen

**Affiliations:** 1Department of Oto-Rhino-Laryngology, Head- and Neck Surgery and Audiology Copenhagen University Hospital—Rigshospitalet, 2100 Copenhagen, Denmark; christoffer.holst.hahn.01@regionh.dk (C.H.H.); fatemeh.makouei@regionh.dk (F.M.); irene.wessel.01@regionh.dk (I.W.); tobias.todsen@regionh.dk (T.T.); 2Institute of Clinical Medicine, Faculty of Health Sciences, Copenhagen University, 2200 Copenhagen, Denmark; 3Department of Radiology, Copenhagen University Hospital—Rigshospitalet, 2100 Copenhagen, Denmark; martin.lundsgaard.hansen@regionh.dk; 4Department of Pathology, Copenhagen University Hospital—Rigshospitalet, 2100 Copenhagen, Denmark; tina.klitmoeller.agander@regionh.dk; 5Copenhagen Academy for Medical Education and Simulation, Capital Region, 2100 Copenhagen, Denmark

**Keywords:** oral cancer, tongue cancer, staging, depth of invasion, intraoral ultrasound, MRI, pathology

## Abstract

**Simple Summary:**

Oral squamous cell carcinoma of the tongue is the most common type of oral cavity cancer. The depth of tumor invasion is an important factor for both treatment planning and prognosis. In this study, we investigated the accuracy of intraoral ultrasound and magnetic resonance imaging (MRI) in assessing the depth of invasion in patients with oral tongue squamous cell carcinoma. Histopathological measurement of DOI was used as a reference standard. We conducted a prospective study including 30 patients planned for surgical treatment of oral tongue squamous cell carcinoma. Statistical analysis showed that intraoral ultrasound was more accurate than MRI for assessing the depth of invasion of oral tongue cancers. Clinical practice and guidelines could implement intraoral ultrasound accordingly.

**Abstract:**

Oral squamous cell carcinoma (OSCC) of the tongue is the most common type of oral cavity cancer, and tumor depth of invasion (DOI) is an important prognostic factor. In this study, we investigated the accuracy of intraoral ultrasound and magnetic resonance imaging (MRI) for assessing DOI in patients with OSCC. Histopathological measurement of DOI was used as a reference standard. We conducted a prospective study including patients planned for surgical treatment of OSCC in the tongue. The DOI was measured in an outpatient setting by intraoral ultrasound and MRI, and was compared to the histopathological DOI measurements. Bland–Altman analysis compared the mean difference and 95% limits of agreement (LOA) for ultrasound and MRI, and the Wilcoxon signed-rank test was used to test for significance. The correlation was evaluated using Pearson’s correlation coefficient. We included 30 patients: 26 with T1 or T2 tumors, and 4 with T3 tumors. The mean difference from histopathology DOI was significantly lower for ultrasound compared to MRI (0.95 mm [95% LOA −4.15 mm to 6.06 mm] vs. 1.90 mm [95% LOA −9.02 mm and 12.81 mm], *p* = 0.023). Ultrasound also led to significantly more correct T-stage classifications in 86.7% (26) of patients compared to 56.7% (17) for MRI, *p* = 0.015. The Pearson correlation between MRI and histopathology was 0.57 (*p* < 0.001) and the correlation between ultrasound and histopathology was 0.86 (*p* < 0.001). This prospective study found that intraoral ultrasound is more accurate than MRI in assessing DOI and for the T-staging of oral tongue cancers. Clinical practice and guidelines should implement intraoral ultrasound accordingly.

## 1. Introduction

Oral squamous cell carcinoma (SCC) of the tongue is the most common type of oral cavity cancer, with less than 60% of patients surviving more than 5 years [1]. Surgery is recommended as a single treatment for early stage T1 and T2 oral tumors, while surgery followed by radiotherapy or chemoradiotherapy should be considered in advanced stages (T3 and T4) [2,3].

The depth of invasion (DOI) of oral tongue SCC is an important prognostic factor in predicting lymph node metastasis and survival, and has been included in the latest Tumor, Node and Metastasis (TNM) classification by the American Joint Committee on Cancer [4,5,6,7]. A tumor with a DOI between 5 and 10 mm is classified as at least T2, and a tumor with a depth of invasion of more than 10 mm is classified as at least T3 [7,8,9,10]. Sentinel node biopsy is a treatment option in some centers for neck management in early stage (T1–T2) tumors with clinical N0 neck [2,3]. Another treatment strategy is to perform elective neck dissection if there is a DOI of more than 4 mm. [11]. Accurate assessment of the DOI in tongue tumors is therefore important to correctly classify the T-stage and determine the optimal cancer treatment. Further, an accurate preoperative DOI measurement may also help the surgeon estimate the amount of tissue that needs to be resected.

Magnetic resonance imaging (MRI) is widely used as a standard preoperative imaging modality but has difficulty assessing DOI in smaller tumors. In some cases of small tumors, the tumor is not recognizable on the MRI scan, and therefore the DOI cannot be assessed. These are, therefore, a challenge in clinical practice as the DOI will determine which cases should be managed with sentinel node biopsy rather than standard neck dissection [11,12,13,14]. Instead, the development of small-footprint ultrasound transducers has made it possible to directly scan the tongue tumor surface with high-frequency intraoral ultrasound. There has been an increasing number of studies on this recently, that have found promising results using intraoral ultrasound for DOI measurements, even in larger series. There are studies in which the tumor thickness was measured, but also several studies where DOI has been measured, often with good results [11,15,16,17,18]. However, due to the limited evidence, MRI is still recommended as the first choice image modality for T-staging [7]. Therefore, to support the use of intraoral ultrasound in the diagnostic workup of tongue cancer patients, this prospective study aimed to compare the accuracy of DOI measurements by intraoral ultrasound and MRI of squamous cell carcinoma of the tongue.

## 2. Materials and Methods

We conducted a prospective diagnostic accuracy study at the Department of Oto-rhinolaryngology, Head and Neck Surgery, and Audiology, Rigshospitalet, University of Copenhagen, Copenhagen, Denmark. Patients were given oral and written information and signed written consent before enrolment in the study. All patients received their cancer treatment free of charge in a tax-funded public healthcare system. The protocol was registered at “ClinicalTrials.gov/study/NCT04614896 (accessed on 15 October 2020)” (ID NCT04614896) and reported to the Regional Ethics Committee of the Capital Region of Denmark, which considered it exempt from further processing (ID H-20033152).

Eligible participants were patients with squamous cell carcinoma of the oral tongue, planned for surgery at the Department of Otorhinolaryngology, Head and Neck Surgery, and Audiology, Rigshospitalet—a tertiary referral hospital covering the surgical cancer treatment for a population of 2.6 million. The patients were invited to participate in the study before or at the multidisciplinary team conference where the treatment was planned, if one of the surgeons performing the intraoral ultrasound examination was available. A histopathological diagnosis of squamous cell carcinoma was required for inclusion. We included only patients with a tongue tumor and excluded patients with a tumor growing into adjacent tissue. We also excluded patients previously treated for tongue SCC, but not patients treated for other cancers. All patients enrolled had an intraoral ultrasound and MRI performed blinded to each other before their surgical cancer treatment. The patients were included between December 2020 and November 2022.

The study was performed as a diagnostic accuracy study, with measurements of depth of invasion on intraoral ultrasound and MRI as the two index tests, and measurement of depth of invasion on histopathological examination as the reference test. The reporting was conducted following the STARD guidelines [19].

### 2.1. Intraoral Ultrasound Examination

The intraoral ultrasound examination was performed in the outpatient clinic or the operating theater under general anesthesia before surgery. The ultrasound exam was conducted with an 18 MHz “Hockey Stick” X18L5s and 18 MHz linear 18L5 transducer or with a 15 MHz Hockey Stick 8809 transducer and 18 MHz linear 8870 transducer, depending on which ultrasound system was available, either BK3000 or Flex Focus 400 (both BK Medical^®^, Burlington, MA, USA), respectively. The image optimization mainly included the depth, frequency, gain, and focus adjustment to ensure the minimum required depth, the highest possible resolution and contrast, and the focus point approximately at the center of the tumor region.

All intraoral ultrasound scans were performed by one of two head and neck surgeons (MK and CHH), both with more than 15 years of experience with head and neck ultrasound. An ultrasound probe cover was used, and the tongue was gently held with a gauze, while the ultrasound probe was placed directly on the tumor (see Figure 1). A sweep was performed in the sagittal and transversal planes to determine the deepest part of the tumor.

Using ultrasound, DOI was estimated either by the well-defined border of the hypoechoic tumor or, if the border was irregular, the deepest part of the hypoechoic area (see Figure 2).

To compensate for the deeper invasion of ulcerative tumors and the more superficial invasion of mass-formed tumors, the assessment of DOI aimed to measure the depth from the mucosal border to the deepest part of the tumor (see Figure 3). If the tumor could not be visualized using ultrasound, the depth was registered as 0 mm. All ultrasound images were stored for documentation.

#### MRI Examination

MRI of the head and neck were performed on 1.5T and 3.0T systems using standard ENT protocols. Available sequences were T1 with and without contrast, T2, and Short-TI Inversion Recovery (STIR) imaging, and the slice thickness was between 4 and 6 mm. The MRI data were evaluated by the same head and neck radiologist (ML) and measurements were carried out upon visibility on the axial, coronal, and sagittal planes. An average value was calculated if the depth of invasion (DOI) was visible on two planes. In cases where DOI could not be determined on MRI, the depths were set to 0 mm for analysis. Images were viewed in the PACS system (IMPAX 6, AGFA).

### 2.2. Histopathology Measurements

DOI and TT were measured according to current recommendations on histopathological examination from the American Joint Committee on Cancer and the College of American Pathologists [7,20,21]. DOI is measured in mm from the level of the basement membrane of the closest adjacent normal mucosa perpendicularly to the deepest point of tumor invasion, while TT is the distance from the mucosal surface of the tumor to the deepest point of invasion. All histopathologic examinations were performed by an experienced head and neck pathologist. The pathologists were blinded from both ultrasound and MRI measurements before the histopathological examination.

### 2.3. Statistics

#### Power Calculation

We assumed to find a mean 2 mm difference between ultrasound and MRI compared to the histopathology DOI, with a standard deviation of 2.5 mm. We, therefore, needed 25 patients to obtain a power of 0.8 and α 0.05. We therefore aimed to include 30 patients in the study.

The degree of correlation was evaluated using Pearson’s correlation coefficient for intraoral ultrasound and MRI, with histological DOI measurements used as reference standard. Bland–Altman plots were used to evaluate the mean difference between the index tests (intraoral ultrasound and MRI) and the reference standard (histopathological DOI measurements), with a 95% confidence interval. The Wilcoxon signed-rank test was used to test if there was a significant difference between the two index tests. R programming language and the ggplot2 package [22] were used for statistical analysis and a two-side 5% significance level was applied.

## 3. Results

A total of 32 patients with T1–T3 tongue SCC were enrolled in the study from December 2020 to November 2022. Two patients were excluded; one patient had radiation instead of surgery, and another had a tumor infiltrating the floor of the mouth (See Figure 4 for STARD flow diagram).

The demographic and clinical data are summarized in Table 1. The mean age of the included patients was 68.1 years (median 68.9 years) and the range was 40 to 86 years, standard deviation 10.6 years. Sixty percent were men. The majority of the tumors (86%) were T1 or T2 tumors, and 70% were clinically N0.

We included 30 patients for the final analyses; all patients had an intraoral ultrasound performed, and all patients had an MRI scan performed. No adverse events were seen in relation to the index tests. DOI could be measured in 30/30 cases using ultrasound, while using MRI, the DOI could not be measured in 7 patients and was, therefore, set to 0 mm. All patients were operated on within 14 days of the intraoral ultrasound and MRI scan, hence this is the maximum interval between the index tests and the reference test.

Figure 5 shows two examples of clinical images, ultrasound images, MRI images, and histopathological images of different tongue carcinomas. The first is an exophytic tumor, where the DOI was measured to be 5.85 mm using MRI, 4 mm using intraoral ultrasound, and the DOI was measured to be 3 mm upon histopathological examination. This is an example of a case where the tumor thickness is larger than the DOI. The second case shows a more ulcerative tumor where the DOI was measured to be 5.8 mm using MRI, 5.5 mm using intraoral ultrasound, and 5.5 mm upon histopathological examination. In this case, the tumor thickness is similar to the DOI.

The mean DOIs were measured to be 7.3 mm (SD 6.7), 6.4 mm (SD 5.2), and 5.4 mm (SD 4.7) for MRI, US, and histopathological examination, respectively.

The mean absolute difference between DOI measured using MRI and histopathology was 3.70 mm (95% limits of agreement (LOA) −9.0–12.8 mm), and the Pearson correlation was 0.57, which indicates a moderate correlation. The mean absolute difference between histopathologic DOI and intraoral ultrasound DOI was 1.63 mm (95% LOA −4.1–6.1 mm), and the Pearson correlation was 0.86, which indicates a very strong correlation. Both intraoral ultrasound and MRI showed a significant correlation to histopathology, *p* < 0.001. (See Table 2 and Figure 6)

When performing the Wilcoxon signed-rank test, comparing the absolute difference between DOI measurement by intraoral ultrasound and pathology, and MRI and pathology, intraoral ultrasound performed better in 22 cases, making it significantly better (*p* = 0.023).

Ultrasound was significantly more accurate for T-staging than MRI, as DOI measurements led to a correct T-stage classification in 26 (86.7%) and 17 (56.7%) of the patients, for intraoral ultrasound and MRI, respectively (*p* = 0.015), see Table 3.

Eight patients had N+ disease, and their mean pathological DOI was 9.38 mm (range 3–18 mm). This was significantly larger than in patients with N0 disease, who had a mean DOI of 4.07 mm (range 0.5–18 mm), *p* = 0.02. The ranges for the intraoral ultrasound and MRI measurements of DOI in N+ patients were 2.4 to 18.2 mm and 0–15.3 mm, respectively.

## 4. Discussion

This prospective study found that ultrasound was significantly more accurate than MRI in measuring the DOI and determining the clinical T-staging of oral tongue squamous cell carcinoma.

The narrower limits of agreement also showed a lower ultrasound variation than MRI compared to postoperative histopathological measurements.

A strength is that our study is one of only two prospective studies assessing the accuracy of intraoral ultrasound and MRI for determining the DOI [16,23], and the first study to show that intraoral ultrasound performs significantly better than MRI when measuring the DOI and determining T-staging [23]. Another strength of our study is that the same two surgeons assessed all the patients included in the study with a standardized scanning approach to limit the operator dependency of ultrasound. Unlike other studies [24], we used no special equipment for the intraoral ultrasound except for a transducer cover, making it easy for the surgeon to perform in the outpatient clinic, and easier for other head and neck centers to reproduce. A limitation of our study is the rather small sample size. This limited the possibility to perform subgroup analysis. However, based on the power calculation, we included the planned number of patients, which could show that intraoral ultrasound performed better than MRI. We also believe that these significant study results can be generalized to other head and neck cancer clinics.

Our findings are comparable to previous studies that reported correlations between 0.8 and 0.98 between ultrasound and histologic measurements of DOI or TT [15]. Lida et al. found a correlation of 0.87 between ultrasound and histopathology, and Takamura found a correlation of 0.83 [24,25]. These values are comparable to the correlation of 0.86 that we found.

Our study finds MRI and ultrasound to overestimate the DOI by 1.90 and 0.95 mm, respectively. An overestimation of the DOI can also increase the T-stage classification, thereby changing the treatment. It is known that about 10% shrinkage will happen after the excision and formalin fixation of the tissue [26,27,28], and this shrinkage could explain part of the overestimation of the DOI, but not all. Further, the shrinkage of the surgical specimen does not explain why MRI overestimates DOI more than intraoral ultrasound. A possible explanation for this could be that it is more difficult using MRI to distinguish between tumor and edema in the adjacent tissue.

Measurement of the estimated DOI was possible on all 30 patients (100%) with ultrasound, and only in 23 patients (77%) with MRI. When the DOI could not be measured using MRI, the histopathological DOI was between 0.5 and 3 mm. The mean DOI was significantly higher for patients with cervical lymph node metastasis than N0 patients, 9.38 mm vs. 4.07 mm, respectively. The ranges of the groups were overlapping and, therefore, do not support a specific cut off value of DOI to rule out the spread to lymph nodes.

The AJCC guidelines recommend CT and MRI for T-stage evaluation of oral tongue squamous cell carcinoma [7]. Ultrasound may be a useful adjunct for estimating the DOI but is still not recommended as the primary imaging modality, as more evidence is needed [24]. Our study demonstrates that ultrasound is more accurate than MRI in determining DOI and leads to better T-stage classification. As intraoral ultrasound can easily be performed in the outpatient clinic, we believe it should be recommended as additional imaging for DOI measurements in patients where it may change the T-stage and choice of treatment. Ultrasound is also a portable and dynamic image modality, and the visualization of DOI may also be useful as preoperative guidance for the surgeon to improve surgical outcomes. Further, ultrasound may also be useful as intraoperative imaging of resection margins, as described in a few studies [29,30,31], and which needs to be explored in future studies. Also, the use of 3D ultrasound in determining surgical margins in oral cancer surgery is an area of interest [32], as is the use of transoral ultrasound for the diagnostic workup of oropharyngeal cancers [33].

## 5. Conclusions

Our prospective clinical study shows that intraoral ultrasound provides a more accurate preoperative assessment of DOI than MRI and can significantly improve the clinical T-staging of oral tongue squamous cell carcinoma. We believe that intraoral ultrasound and assessment of these tumors is possible to perform in other centers with experience in head and neck ultrasound, and that our results are reproducible. We thus believe that intraoral ultrasound should be an integrated part of the diagnostic workup of patients diagnosed with oral tongue cancer to ensure correct tumor classification and treatment planning, and that this modality should be implemented in clinical staging guidelines.

## Figures and Tables

**Figure 1 cancers-16-00637-f001:**
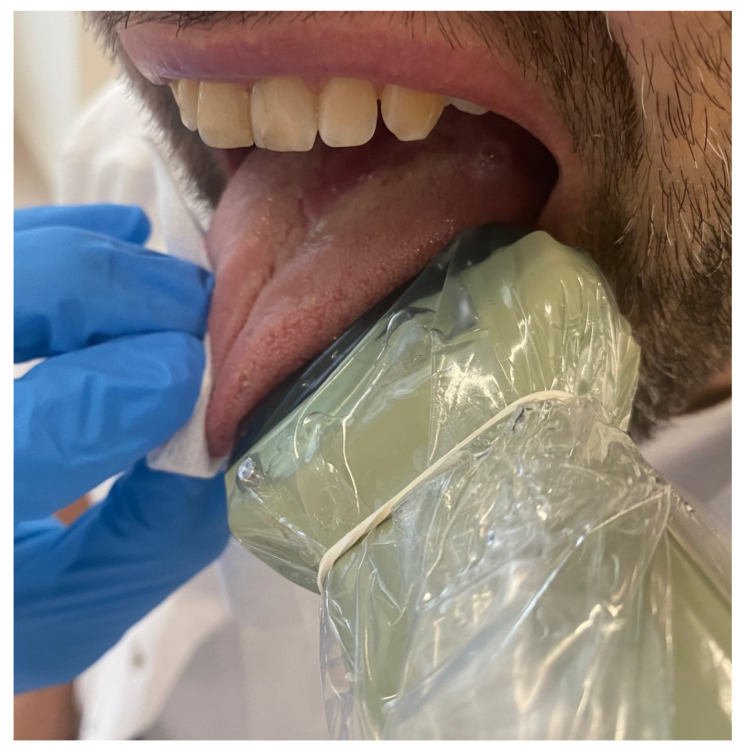
How the intraoral ultrasound examination is performed in the outpatient clinic. The tongue is held with a gauze, and the transducer is placed directly on the tumor.

**Figure 2 cancers-16-00637-f002:**
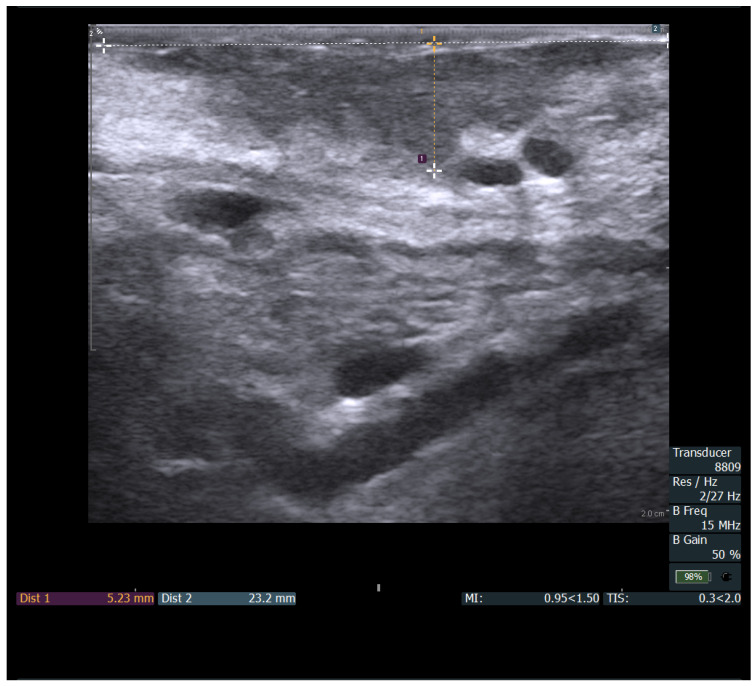
Ultrasound image of a tongue tumor. The horizontal line is used to estimate the level of the basal membrane, and the vertical line is the estimated depth of invasion.

**Figure 3 cancers-16-00637-f003:**
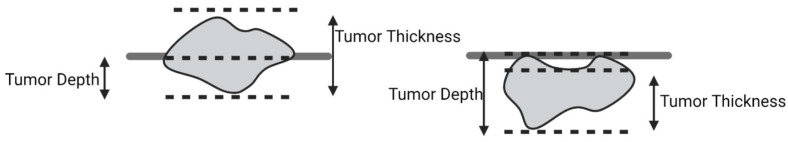
Tumor thickness can be both larger and smaller than the depth of invasion. This must be taken into account when performing the intraoral ultrasound.

**Figure 4 cancers-16-00637-f004:**
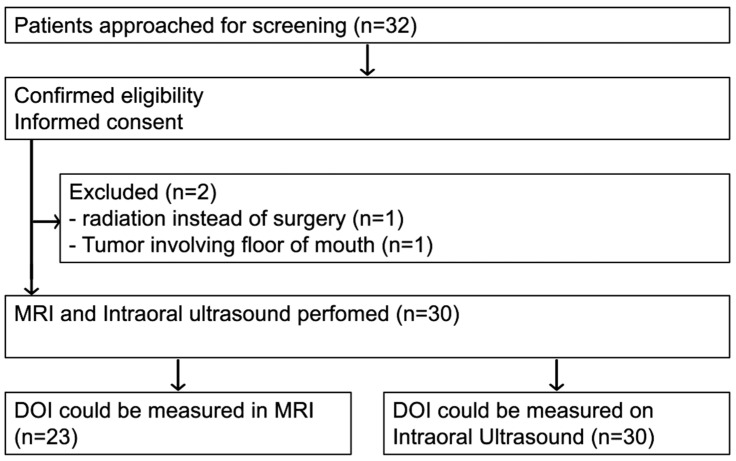
Flow diagram of patients in the study.

**Figure 5 cancers-16-00637-f005:**
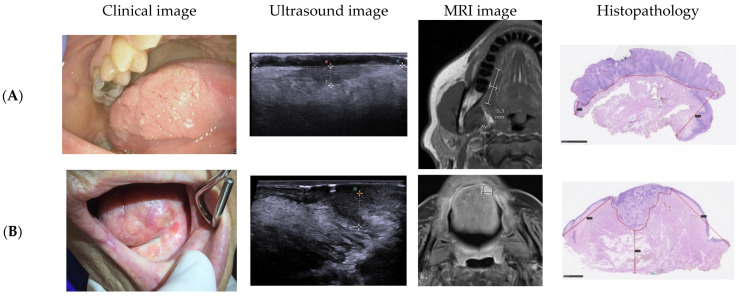
Examples of different clinical presentations, ultrasound, MRI, and histopathological images. (**A**). Exophytic tumor; (**B**). Ulcerative tumor.

**Figure 6 cancers-16-00637-f006:**
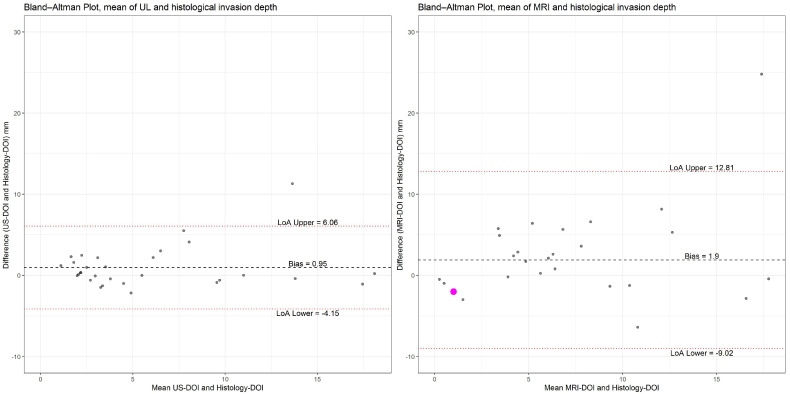
Bland–Altman analysis of US-DOI versus histopathology-DOI. n = 30 and of MRI-DOI versus histopathology-DOI. n = 30. The dashed gray line represents the mean difference and the dotted red lines represent 95% limits of agreement (LOA). On the left *X*-axis: Mean US-DOI and histopathology-DOI, *Y*-axis: difference (US-DOI—HISTOPATH-DOI) mm. On the right *X*-axis: Mean MRI-DOI and histopathology-DOI, *Y*-axis: difference (MRI-DOI—HISTOPATH-DOI) mm. Purple dot represents five separate entries in the plot.

**Table 1 cancers-16-00637-t001:** Patient demographics and clinical information.

Patient Characteristics	n = 30
Median age (range)	69 (40–86)
Male	18 (60%)
Female	12 (40%)
Smoking	
Current	4 (13%)
Earlier	19 (63%)
Never	6 (20%)
No data	1 (3%)
Alcohol consumption	
Yes	22 (73%)
No	7 (23%)
No data	1 (3%)
Tumor characteristics	n = 30
T stage	
T1	16 (53%)
T2	10 (33%)
T3	4 (13%)
Cohesive front	2 (7)
Non-cohesive front	23 (77)
No information on cohesivity	5 (20)
Perineural growth	15 (50)
No perineural growth	13 (43)
No information on Perineural growth	2 (7)
N-stage	
N0	21 (70)
N1	4 (13)
N2	2 (7)
N3	2 (7)
n/a	1 (3)

**Table 2 cancers-16-00637-t002:** Results of the two index tests.

	Test Method (Index Test)	Histopathology (Reference Test)				
	Mean (SD)	Mean (SD)	Mean Absolute Difference	Lower LOA	Upper LOA	Pearson Correlation
US vs. hist	6.4 (5.2)	5.4 (4.7)	1.6	−4.1	6.1	0.86
MRI vs. hist	7.3 (6.7)	5.4 (4.7)	3.7	−9.0	12.8	0.57

LOA = Limits of agreement.

**Table 3 cancers-16-00637-t003:** T-stage accuracy for US and MRI.

Modality	T-Stage Overestimated	T-Stage Underestimated	T-Stage Correctly Estimated
Intraoral Ultrasound	4	0	26
MRI	11	2	17

## Data Availability

The data presented in this study are available in this article.

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
