# Peer review of "Intraoral Ultrasound versus MRI for Depth of Invasion Measurement in Oral Tongue Squamous Cell Carcinoma: A Prospective Diagnostic Accuracy Study"

_cancers, 2024, doi:10.3390/cancers16030637_

Round 1
Reviewer 1 Report
Comments and Suggestions for Authors
Dear Authors, thank you for your paper. The study is about a topic really interesting and with good impact in the scientific community. Also, the US use in such OSCCs surely will be widespread in the next times, so paper as yours reporting on such issue are importan for the scientific community. Some concerns: your study is on a relative low number of patients but nevertheless is importan, for such reason your work must be improved with pictures especially with comparative picture among clinic, histological low power magnificATION ON E.E., MRI images and US images, both for a small OSCC and for a larger and also for OSCC with different clinical presentation (nodule or ulcer); this considering that US use is strictly operator related and the visual impact is the key for its use, especially in comparison to MRI. Also, expand list of citation as several papers with larger number of patients exist.
Thank you again for your paper
Author Response
Response to reviewer 1:
Dear Authors, thank you for your paper. The study is about a topic really interesting and with good impact in the scientific community. Also, the US use in such OSCCs surely will be widespread in the next times, so paper as yours reporting on such issue are importan for the scientific community. Some concerns: your study is on a relative low number of patients but nevertheless is importan, for such reason your work must be improved with pictures especially with comparative picture among clinic, histological low power magnificATION ON E.E., MRI images and US images, both for a small OSCC and for a larger and also for OSCC with different clinical presentation (nodule or ulcer); this considering that US use is strictly operator related and the visual impact is the key for its use, especially in comparison to MRI.
Thank you for your interest in our study and for your suggestions to improve the manuscript. We agree that a study limitation is the small sample size. However, it was still sufficiently powered to demonstrate that the US was significantly better than MRI at evaluating the depth of invasion. Therefore, the study contributes new scientific data to support the development of evidence-based guidelines for staging oral tongue squamous cell carcinoma. We also agree that more pictures could improve the manuscript, and we have added pictures of how the examination is performed (figure 1), clinical photos, ultrasound images, MRI images, and histological images of different tumor types (figure 5).
Also, expand list of citation as several papers with larger number of patients exist.
Thank you for highlighting the missing citations. We agree that there are several publications about intraoral ultrasound, however, most of them evaluate tumor thickness and not the depth of invasion as our study. Still, we agree that it could be beneficial to cite and discuss their findings in our manuscript. We have therefore added 3 citations (refs 11,17 and 18) of other studies in the introduction and discussion section:
“…Instead, the development of small-footprint ultrasound transducers has made it possible to directly scan the tongue tumor surface with high frequency intraoral ultrasound. There has been an increasing number of studies of this recently, that have found promising results using intraoral ultrasound for DOI measurements, even in larger series. There are both studies, in which the tumor thickness was measured, but also several studies where DOI has been measured, often with good results [11,15-18]. However, due to the limited evidence, MRI is still recommended as the first choice image modality for T-staging [7]. “(Page 2, lines 87–94)
Reviewer 2 Report
Comments and Suggestions for Authors
The topic of the manuscript is interesting, and it is always commendable to present clinical results. However, the presentation is not good, the introduction does not introduce the reader to the problem, the methodology is not properly written, does not follow the CONSORT guidelines, nor does it have a flowchart.
In both abstracts, this sentence is irrelevant: We included 30 patients; 60% were male, and the median age was 69 years. - please delete it.
Please, what does this mean: The Pearson correlation to histopathology 36
were 0.57 and 0.86 for MRI and ultrasound, respectively. (besides the r value, the p value is important).
The abstract is poorly written, it is not fluent, please correct it.
What does the abbreviation, TNM classification mean - please state the full meaning. Ditto for, MRI.
Please specify the purpose of the work in more detail at the end of the introduction?
The period in which the study was conducted should be written in the methods, and the same should be written in accordance with the CONSORT guidelines. Transfer the flowchart to the methodology.
Along with the mean age of the patient, put Sd and the median age.
If MRI DOI was measured in 7 patients, then how to compare the results and draw conclusions. You do not have a sufficient number according to the sample size calculation. I think that the study can be conducted on a more comprehensive number of respondents.
Diksuija is short, does not explain the results enough and is not interesting.
What is the strength of the study, if any? There are no limitations listed either.
I am sorry, but I am of the opinion that this study, presented in this way and carried out on an insufficient number of patients, cannot be considered for publication.
Author Response
The topic of the manuscript is interesting, and it is always commendable to present clinical results. However, the presentation is not good, the introduction does not introduce the reader to the problem, the methodology is not properly written, does not follow the CONSORT guidelines, nor does it have a flowchart.
Thank you for the comments and suggestions to improve our manuscript. We have used the STARD guidelines (STARD stands for “Standards for Reporting Diagnostic Accuracy Studies”), which are recommended for diagnostic accuracy studies. As we did not perform a randomized clinical trial, we do not believe the CONSORT (Consolidated Standards of Reporting Trials) guidelines are the most relevant to use in our manuscript. However, we agree that a flowchart will improve the presentation of our data, and we have added this to the manuscript as Figure 4 as recommended in the STARD guidelines.
In both abstracts, this sentence is irrelevant: We included 30 patients; 60% were male, and the median age was 69 years. - please delete it.
Thank you for this comment. We have now edited the text in the abstract based on your suggestion:
“We included 30 patients, 26 with T1 or T2 tumors, and 4 with T3 tumors.” (Page 1, line 32)
Please, what does this mean: The Pearson correlation to histopathology were 0.57 and 0.86 for MRI and ultrasound, respectively. (besides the r value, the p value is important).
Thank you for this comment and the opportunity to improve the presentation of our data. We have now added the p-values to the abstract and rewritten the text, see lines 36-38: “The Pearson correlation between MRI and histopathology was 0.57 (p < 0.001) and the correlation between ultrasound and histopathology was 0.86 (p < 0.001).”
The abstract is poorly written, it is not fluent, please correct it.
We have made some changes, including the above-mentioned, to make it more fluent.
What does the abbreviation, TNM classification mean - please state the full meaning. Ditto for, MRI.
Thank you for noticing this. We have added “Magnetic Resonance Imaging” (MRI) and “Tumor, Node and Metastasis” (TNM) in full typing. Lines 16, 24, 51 and 62.
Please specify the purpose of the work in more detail at the end of the introduction?
Thank you for this suggestion. We have now added some text at the end of the introduction to explain the purpose more clearly:
“However, due to the limited evidence, MRI is still recommended as the first choice image modality for T-staging [7]. Therefore, to support the use of intraoral ultrasound in the diagnostic work-up of tongue cancer patients, this prospective study, aimed to compare the accuracy of DOI measurements by intraoral ultrasound and MRI of squamous cell carcinoma of the tongue.” (Page 2, lines 92–97)
The period in which the study was conducted should be written in the methods, and the same should be written in accordance with the CONSORT guidelines. Transfer the flowchart to the methodology.
Thank you for the suggestion to improve the manuscript. We have now moved the study inclusion period in the abstract and method section as suggested. As mentioned earlier, we have are following the STARD guidelines, and according to these, the flowchart of patients should be in the results section.
Along with the mean age of the patient, put Sd and the median age.
Thank you for suggesting. The SD has now been added to the results section:
“The mean age of the included patients was 68.1 years (median 68.9 years) and the range was 40 to 86 years, standard deviation 10,6 years. Sixty percent were men.” (Page 5, lines 217–219)
If MRI DOI was measured in 7 patients, then how to compare the results and draw conclusions. You do not have a sufficient number according to the sample size calculation. I think that the study can be conducted on a more comprehensive number of respondents.
As described in the results section, an MRI was performed on all 30 patients. In 23 cases, it was possible to measure the DOI on MRI, while the tumor was too small to measure the DOI on MRI in 7 cases. Therefore, the radiologist reported 0 mm as MRI DOI measurement in these 7 cases.
Diksuija is short, does not explain the results enough and is not interesting.
What is the strength of the study, if any? There are no limitations listed either.
Thank you for this suggestion to improve the discussion section. We have added more text and references to the discussion part to discuss our results and compare it to related studies:
“A strength is that our study is one of only two prospective studies assessing the accuracy of intraoral ultrasound and MRI for determining DOI [16,22], and the first study to show that intraoral ultrasound performs significantly better than MRI when measuring DOI and determine T-staging. [22] Another strength of our study is that the same two surgeons assessed all the patients included in the study with a standardized scanning approach to limit the operator dependency of ultrasound. Unlike other studies [23], we used no special equipment for the intraoral ultrasound except a transducer cover, making it easy to perform by the surgeon in the outpatient clinic, and easier to reproduce by other Head and Neck centres.” (Page 9, lines 299–307)
Further, we have also elaborated more on some of the limitations of our study:
“A limitation of our study is the rather small sample size. This limited to possibility to perform subgroup analysis. However, based on the power calculation, we included the planned number of patients, could show that Intraoral Ultrasound performed better than MRI. We also believe that these significant study results can be generalized to other head and neck cancer clinics.” (Page 9-10, lines 307–313)
I am sorry, but I am of the opinion that this study, presented in this way and carried out on an insufficient number of patients, cannot be considered for publication.
We hope the manuscript changes, has improved its clarity and clears misunderstandings about the number of patients having MRIs performed.
Round 2
Reviewer 1 Report
Comments and Suggestions for Authors
Dear Authors my suggestions have been satisfied accordingly
Author Response
Dear Reviewer
Thank you for your comments. We are happy that you are satisfied with the changes.
Best regards
Reviewer 2 Report
Comments and Suggestions for Authors
I ask the authors not to write we in the scientific manuscript, but in this study and ect.
I also ask that a new flowchart be made following the consort guidelines.
I also ask that they correct the English language so that the manuscript has a nicer flow.
Comments on the Quality of English LanguageI also ask that they correct the English language so that the manuscript has a nicer flow. The way of writing the manuscript itself is quite confusing.
Author Response
Dear reviewer
Thank you for your comments and suggestions for improving our manuscript. We had English editing to improve the language and grammar of the manuscript.
We have used the STARD guidelines for reporting our results as recommended by the EQUATOR (Enhancing the QUAlity and Transparency Of health Research) Network *. According to these recommendations STARD guidelines should be used for diagnostic accuracy studies like ours while the CONSORT guidelines are used for randomized controlled studies. To increase the transparency of our reporting, we have added the STARD checklist.
Because of the above, the flowchart is made according to the STARD guidelines.
* https://www.equator-network.org/
We hope these changes will meet your expectations.
Best regards
Round 3
Reviewer 2 Report
Comments and Suggestions for Authors
Congrats